# Integration of LC-MS-Based and GC-MS-Based Metabolic Profiling to Reveal the Effects of Domestication and Boiling on the Composition of Duck Egg Yolks

**DOI:** 10.3390/metabo13010135

**Published:** 2023-01-16

**Authors:** Yong Tian, Guoqin Li, Xizhong Du, Tao Zeng, Li Chen, Wenwu Xu, Tiantian Gu, Zhengrong Tao, Lizhi Lu

**Affiliations:** 1State Key Laboratory for Managing Biotic and Chemical Threats to the Quality and Safety of Agro-Products, Institute of Animal Science & Veterinary, Zhejiang Academy of Agricultural Sciences, Hangzhou 310021, China; 2Key Laboratory of Livestock and Poultry Resources (Poultry) Evaluation and Utilization, Ministry of Agriculture and Rural Affairs of China, Hangzhou 310021, China; 3Institute of Animal Husbandry and Veterinary Medicine, Jinhua Academy of Agricultural Sciences, Jinhua 321017, China

**Keywords:** duck egg yolk, metabolomics, domestication, boiling, lipid metabolism, stress resistance

## Abstract

Egg yolks contain abundant lipids, proteins, and minerals that provide not only essential nutrients for embryonic development but also cheap sources of nutrients for consumers worldwide. Previous composition analyses of egg yolks primarily focused on nutrients such as lipids and minerals. However, few studies have reported the effects of domestication and heating on yolk composition and characteristics. The objective of this study was to investigate the impact of domestication and boiling on the metabolite contents of egg yolks via untargeted metabolomics using GC-MS and LC-MS. In this study, eggs were collected from Fenghua teals, captive mallards, and Shaoxing ducks. Twelve duck eggs (half raw and half cooked) were randomly selected from each variety, and the egg yolks were separated for metabolic profiling. The analysis identified 1205 compounds in the egg yolks. Domestication generated more differential metabolites than boiling, which indicated that the changes in the metabolome of duck egg yolk caused by domestication were greater than those caused by boiling. In a comparative analysis of domestic and mallard ducks, 48 overlapping differential metabolites were discovered. Among them, nine metabolites were upregulated in domesticated ducks, including monoolein, emodin, daidzein, genistein, and glycitein, which may be involved in lipid metabolism; some of them may also act as phytoestrogens (flavonoids). Another 39 metabolites, including imethylethanolamine, harmalan, mannitol, nornicotine, linoleic acid, diphenylamine, proline betaine, alloxanthin, and resolvin d1, were downregulated by domestication and were linked to immunity, anti-inflammatory, antibacterial, and antioxidant properties. Furthermore, four overlapping differential metabolites that included amino acids and dipeptides were discovered in paired comparisons of the raw and boiled samples. Our findings provided new insights into the molecular response of duck domestication and supported the use of metabolomics to examine the impact of boiling on the composition of egg yolks.

## 1. Introduction

Avian eggs are one of the most important sources of nourishment for humans. According to statistics, the global annual production of duck eggs is about 4.2 million tons. They are highly valued by consumers worldwide, particularly in East and Southeast Asia [1] where duck eggs account for more than 90% of the total consumption. Egg yolk is rich in lipids, amino acids, proteins, and other nutrients, particularly phospholipids and choline, which are beneficial to health. The yolk of a bird offers the major structural and energetic components required for the development of the embryo protected within the eggshell as an excellent supply of high-quality proteins, lipids, and numerous bioactive compounds [2]. Previous studies have demonstrated that the nutritional profile of eggs may be altered by the production technique [3], heating [4,5], diet, and breeds [5,6]. Genetics, variation, and selection are well recognized as the three fundamental processes in biological evolution and new variety breeding, and domestication can create new genetic resources. Therefore, we hypothesized that domestication may also affect duck egg yolk composition. To our knowledge, most livestock domestication research has focused on modifications in animal genomes. Information regarding the changes in the composition of avian egg yolks in response to domestication is very limited. Analyzing the influence of domestication on the chemical composition of duck egg yolks will certainly provide important insights into the biological mechanisms involved in the domestication of ducks. Moreover, high-molecular-weight components such as proteins produce additional free amino acids (FAAs) during the heating process. This is important to the understanding of the nutritional properties of food. However, there are no reports related to the effects of boiling on the metabolomic properties of duck egg yolks.

Metabolomics, which is the study of the small molecular metabolites of an organism or cell in a specific physiological state, is used to identify novel metabolites or changes in the ratio of metabolites [7]. Metabolomics can be applied in various fields such as health [8,9,10], plant [11,12], environmental [13,14], and food and nutrition sciences [15,16,17]. Depending on the research purpose, it can be divided into untargeted and targeted metabolomics. Untargeted metabolomics, which focuses on the qualitative detection of untargeted metabolites to provide an overview of the metabolic profile, is usually performed for rapid metabolite screening and determination. Meanwhile, targeted analysis allows for the quantitative analysis of the target compounds. The most exhaustive detection is generally achieved via analytical platforms that include liquid chromatography–mass spectrometry (LC-MS), gas chromatography–mass spectrometry (GC-MS), and nuclear magnetic resonance (NMR) [18]. LC-MS and GC-MS are well-known chromatographic mass spectrometry systems that are sensitive and complimentary. LC-MS is better for detecting lipids, while GC-MS is better for identifying low-molecular-weight compounds. LC-MS and GC-MS metabolome analyses have been used to identify changes caused by diet in bee metabolite profiles [19]. Therefore, metabolomics technology based on these two platforms might provide valuable insights into the impact of domestication and boiling on egg yolk composition.

In this work, we used an untargeted metabolomics technique based on LC-MS and GC-MS to evaluate the metabolic responses of duck egg yolks after domestication and boiling. The duck eggs used in this experiment were produced by Fenghua teals, captive mallards, and Shaoxing ducks. The Fenghua teal is a mallard species that is a local genetic resource formed by long-term natural selection under the unique ecological environment in the offshore area of Fenghua, Zhejiang Province, China. The captive wild ducks used in this study also belong to a mallard species and therefore are called captive mallards. As an excellent egg-laying duck breed in China, the Shaoxing duck is a highly domesticated duck that originated from mallard species [20]. The wild duck, also known as the mallard duck, is the progenitor of all ducks except for the Muscovy duck and is the primary target of artificial domestication. Meanwhile, the Shaoxing duck is an excellent domesticated duck breed with high egg production in China, and its characteristics have been comprehensively researched. However, the understanding of its cell biology or metabolic profile is still limited. This study aimed to reveal the changes in the metabolic profiles of duck egg yolks induced by domestication and boiling and to provide an alternative research method for evaluating their biological effects.

## 2. Materials and Methods

### 2.1. Egg Preparation

The eggs used in this study were collected from three duck varieties. The Fenghua teal eggs (n = 50) and captive mallard eggs (n = 50) were kindly supplied by Ningbo Fenghua Aoji Agricultural Technology Co., Ltd. (Ningbo, Zhejiang Province, China). The eggs from Shaoxing ducks (n = 50) were provided by Jinhua Jinwu Agricultural Development Co. Ltd. (Jinhua, Zhejiang Province, China). All eggs were stored at 4 °C for 48 h until analysis.

### 2.2. Sample Preparation

A total of 36 duck eggs (12 of each variety with a close to average weight) were selected and weighed individually for subsequent analysis. The average weights of the duck eggs (including shells) from the Fenghua teals, captive mallards, and Shaoxing ducks were 57.42 ± 2.78 g, 70.51 ± 4.34 g, and 67.84 ± 4.98 g, respectively. Half of the duck eggs from each breed were collected as raw yolk samples, while the other half were collected after boiling. Eggs from each duck breed were further divided into two groups (raw and cooked) with six eggs per group (n = 6). These groups included raw yolk samples from Fenghua teals (FTR), captive mallards (CMR), and Shaoxing ducks (SCR); and boiled yolk samples from Fenghua teals (FTB), captive mallards (CMB), and Shaoxing ducks (SCB).

The duck eggs were carefully broken to obtain the raw samples, and the whites and yolks were separated manually. The yolks were collected after removing the chalaza, quenched in liquid nitrogen, and stored at −80 °C. A TONZE egg steamer (Guangdong Tonze Electric Co. Ltd., Shantou, Guangdong Province, China) was used to produce the boiled egg yolks. The duck eggs were placed on a plastic plate, and 20 mL of Milli-Q water was filled beneath the plate. The egg steamer was powered on for 10 min and then turned off. The shells were cracked and peeled off, the whites were removed, and the yolks were separated and placed in a 50 mL plastic tube. The samples were quenched with liquid nitrogen and stored at −80 °C.

### 2.3. LC-MS Analysis

Each 50 mg sample was transferred to an EP tube, and 1000 μL of extract solution (methanol:acetonitrile:water = 2:2:1 with the isotopically labeled internal standard mixture) was added. All samples were homogenized at 35 Hz for 4 min and sonicated for 5 min in an ice-water bath. The homogenization and sonication cycle were repeated three times. Then, samples were incubated for 1 h at −40 °C and centrifuged at 12,000 rpm (RCF = 13,800× *g*; r = 8.6 cm) for 15 min at 4 °C. The resulting supernatant was transferred to a fresh glass vial for analysis. The quality control (QC) sample was prepared by mixing an equal aliquot of the supernatants from all samples.

The LC-MS/MS analyses were performed using a Vanquish UHPLC system (Thermo Fisher Scientific, Waltham, MA, USA) with a UPLC BEH Amide column (2.1 mm × 100 mm, 1.7 μm) coupled to a Q-Exactive HFX mass spectrometer (Orbitrap MS, Thermo). Phase A of the liquid chromatography was an aqueous phase containing 25 mmol/L ammonium acetate and 25 mmol/L ammonium hydroxide, while phase B contained acetonitrile. The auto-sampler temperature was 4 °C, and the injection volume was 2 μL. The QE HFX mass spectrometer was employed to acquire the MS/MS spectra in information-dependent acquisition (IDA) mode under the control of the acquisition software (Xcalibur, Thermo). The acquisition software could continuously evaluate the full scan MS spectrum in this mode. The ESI source conditions were set as follows: sheath gas flow rate of 30 Arb, auxiliary gas flow rate of 25 Arb, capillary temperature of 350 °C, full MS resolution of 60,000, MS/MS resolution of 7500, collision energy of 10/30/60 in NCE mode, and spray voltage of 3.6 kV (positive) or −3.2 kV (negative).

### 2.4. GC-MS Analysis

Each 50 mg sample was transferred into a 2 mL EP tube, and steel beads were added. A 1000 μL pre-cooled extraction solution (methanol:acetonitrile:water = 2:2:1) containing an internal standard (2-chloro-L-phenylalanine) was added into the tube. The samples were ground at 35 Hz for 4 min and subjected to ultrasonication in an ice-water bath for 5 min (repeated three times). The treated samples were then stored at −40 °C for 1 h. After centrifugation at 12,000 rpm (RCF = 13,800× *g*; r = 8.6 cm) at 4 °C for 15 min, 400 μL of the supernatant was transferred into a fresh 1.5 mL tube. Each sample of 100 μL was taken out and mixed to prepare the QC sample. After drying the extracts in a vacuum concentrator, 60 μL of methoxyamination hydrochloride (20 mg/mL in pyridine) was added and incubated at 80 °C for 30 min. Then, 80 μL of BSTFA (containing 1% TMCS *v*/*v*) was added to each sample and incubated at 70 °C for 1.5 h. After the mixtures were gradually cooled to room temperature, 5 μL of FAMEs (in chloroform) was added. All samples were then analyzed via gas chromatograph coupled with a time-of-flight mass spectrometer (GC-TOF-MS).

The GC-TOF-MS analysis was performed using an Agilent 7890 gas chromatograph coupled with a time-of-flight mass spectrometer. The system utilized a DB-5MS capillary column (J&W Scientific, Folsom, CA, USA). A 1 μL aliquot of each sample was injected in splitless mode. Helium was used as the carrier gas with a front inlet purge flow of 3 mL min^−1^ and a gas flow rate through the column of 1 mL min^−1^. The initial temperature was kept at 50 °C for 1 min then raised to 310 °C at a rate of 10 °C min^−1^ and held for 8 min at 310 °C. The injection, transfer line, and ion source temperatures were 280 °C, 280 °C, and 250 °C, respectively. The energy was −70 eV in electron impact mode. The mass spectrometry data were acquired in full-scan mode in an *m*/*z* range of 50–500 at a rate of 12.5 spectra per second after a solvent delay of 6.25 min.

### 2.5. Data Analysis

The raw data retrieved from the LC-MS were converted to mzXML format using ProteoWizard software. The peak identification, extraction, alignment, and integration were performed using the R package with an XCMS kernel (Genepioneer Biotechnologies Co., Nanjing, China). Chroma TOF software (V4.3x, LECO, St. Joseph, MI, USA) was used to analyze the initial data from the GC-MS for the peak extraction, baseline adjustment, deconvolution, peak integration, and alignment. All data were merged and pre-processed; i.e., data filtering, processing of missing values, and normalization with Perl program (Genepioneer Biotechnologies Co., Nanjing, China). The filtering criterion included the removal of data without a definite substance name or spectral similarity. Substances with missing values greater than 50% in the samples of each paired comparison were filtered out directly, whereas those with missing values less than 50% were simulated via missing-value recoding using the K-nearest neighbor (KNN) algorithm. The peak area of each metabolite was normalized to the area of the internal standard (IS), and subsequent analyses were conducted based on these normalized values. Identified metabolites were annotated using the KEGG COMPOUND database “https://www.kegg.jp/kegg/compound (accessed on 10 February 2022)”, PubChem database, and Human Metabolome Database (HMDB) V4.0.

A hierarchical clustering analysis (HCA) of the normalized abundance of the metabolome profiles was performed using Origin 2021 (Heat Map with Dendrogram v2.0). SIMCA (V14.1) software was used for the principal component analysis (PCA) and orthogonal partial-least-squares discriminant analysis (OPLS-DA), and the variable importance projection (VIP) values of metabolites were calculated. All metabolites with VIP values greater than 1.0 and *p*-values of the Student’s *t*-test that were less than 0.05 were identified as differential metabolites (DMs) in each paired comparison. Correlations between two groups in each paired comparison were analyzed using GraphPad Prism 9 software. Venn diagrams of metabolites among the comparison groups were constructed using Origin 2021 (Venn Diagram v1.10). A Spearman’s correlation analysis was performed to evaluate the potential links among the metabolites involved in domestication and boiling.

### 2.6. Statistical Analysis

Data regarding the overlapping differential metabolites screened from different comparisons were statistically analyzed via multivariable general linear models using SPSS 26.0 (SPSS Inc., Chicago, IL, USA); a *p*-value of 0.05 was considered statistically significant. The results are presented as the means and the standard errors of the means (SEMs).

## 3. Results

### 3.1. Comprehensive Profiling of Egg Yolk Metabolites

Untargeted metabolomics via UHPLC-MS/MS and GC-TOF-MS analyses was performed to profile the methanol-soluble egg yolk extracts from the FTR, FTB, CMR, CMB, SCR, and SCB groups to evaluate the effects of domestication and boiling on the metabolic changes. The analyses identified 1205 compounds; their profiles were visualized as a clustering heatmap, and then PCA and OPLS-DA scatter plots were constructed. The clustering heatmap (Figure 1A) shows that the domesticated and mallard duck yolks were clustered separately; in the domesticated duck branch, the raw and boiled yolks were clustered together first before merging into a larger branch with the Fenghua teal boiled yolks. In the mallard branch, the boiled yolks of the Fenghua teals and captive mallards were first clustered together and then merged with the raw yolks of the captive mallards.

Egg yolk samples from the six groups of FTR, FTB, CMR, CMB, SCR, and SCB were indistinguishable as shown in the PCA scatter plots (Figure 1B). All samples were within the 95% confidence interval (Hotelling’s T-squared ellipse) and were further analyzed. OPLS-DA was established to validate the metabolic profiles of the yolks in the six groups. As shown in the OPLS-DA score plots (Figure 1C), samples from three duck varieties (two breeds of mallard and one breed of domesticated duck) were completely separated, and differences between the raw and boiled yolks were also evident for each breed. 

The metabolites with VIP values greater than 1.0 and *p*-values less than 0.05 were selected as DMs in each paired comparison to determine the significantly changed metabolites. The DMs were then screened; the results are listed in Appendix A. The number of DMs in paired comparisons of the different groups is shown in Figure 1D. The pairwise comparisons of SCR vs. FTR, SCR vs. CMR, SCB vs. FTB, and SCB vs. CMB between egg yolks of domesticated and mallard ducks revealed 185 (59 upregulated and 126 downregulated), 141 (65 upregulated and 76 downregulated), 199 (70 upregulated and 129 downregulated), and 196 (57 upregulated and 139 downregulated) differential metabolites, respectively. The paired comparisons of FTB vs. FTR, CMB vs. CMR, and SCB vs. SCR between boiled and raw yolks revealed 69 (45 upregulated and 24 downregulated), 85 (75 upregulated and 10 downregulated), and 45 (26 upregulated and 19 downregulated) differential metabolites, respectively. A total of 460 DMs were detected in all paired comparisons, and their dynamic changes were visualized in heatmaps (Figure 1E). The clustering heatmap shows that samples of raw and boiled egg yolks from the three duck breeds were clustered separately. Additionally, samples from the Fenghua teals and captive mallards were clustered together in the clade of mallards.

### 3.2. Changes in Major Metabolites in Paired Comparisons

The major DMs between different paired comparisons were screened based on the lowest *p*-values. Pairwise comparisons were made between the domesticated and wild duck yolk samples (SCR vs. FTR, SCR vs. CMR, SCB vs. FTB, and SCB vs. CMB) and between the boiled and raw yolks (FTB vs. FTR, CMB vs. CMR, and SCB vs. SCR). The results revealed that 2-hydroxymyristic acid, cytosine, imidazole acetol-phosphate, mannitol, undecanoic acid, 12-hydroxydodecanoic acid, pc(18:1(9z)/18:0), daidzein, 3-amino-2-piperidone, and 2-pyrrolidinone were the metabolites with the greatest differences between SCR and FTR (Figure 2A). The relative amount of daidzein in the SCR group was reduced compared to that of the FTR group, while the relative amounts of the remaining nine metabolites increased. The results also revealed that (Â±)-anisoxide, 3′-hydroxy-e,e-caroten-3-one, 3-amino-2-piperidone, pipecolic acid, 2-acetylpyrrolidine, alloxanthin, cytosine, glycitein, 2-propenyl 2-aminobenzoate, and imidazole acetol-phosphate were the top 10 DMs with the lowest *p*-value in a comparison between the SCR and CMR groups; 2-acetylpyrrolidine and glycitein were relatively high in the CMR group, while the remaining 8 metabolites were relatively high in the SCR group (Figure 2B). In the comparison between SCB and FTB, the 10 metabolites that differed the most included dimethylethanolamine, diphenylamine, mannitol, pyrrolidine, 2-pyrrolidinone, lysope(0:0/14:0), pc(20:1(11z)/15:0), ganodosterone, 1h-pyrrole-2-carboxaldehyde, and 2-propenyl 2-aminobenzoate (Figure 2C), which were found to be relatively high in the FTB group. Between the SCB and CMB groups, the top 10 differential metabolites were mannitol, alloxanthin, (Â±)-anisoxide, lysopc(18:2(9z,12z)), lysopc(14:0/0:0), 4-aminobutyraldehyde, lysope(0:0/22:1(13z)), beta-doradecin, imidazole acetol-phosphate, and 2-acetylpyrrolidine, of which 9 metabolites were relatively high in CMB, while 2-acetylpyrrolidine was relatively high in SCB (Figure 2D).

Between the FTR and FTB groups, the 10 metabolites that differed the most were pc(18:2(9z,12z)/15:0), l-cyclo(alanylglycyl), 4-guanidinobutanoic acid, 3-hydroxy-3-(3,4-dihydroxy-4-methylpentanoyl)-5-(3-methylbutyl)-1,2,4-cyclopentanetrione, isoleucyl-leucine, (5r,6s)-5,6-epoxy-7-megastigmen-9-one, leucyl-phenylalanine, pc(16:1(9z)/15:0), pe(18:0/22:6(4z,7z,10z,13z,16z,19z)), and cystine; the relative contents of pc(18:2(9z,12z)/15:0) and pc(16:1(9z)/15:0) were found to be relatively high in the FTR group, and the remaining 8 metabolites were found to be relatively high in the FTB group (Figure 2E). In the comparison between the CMB and CMR groups, the top 10 differential metabolites were l-cyclo(alanylglycyl), lysopc(14:0/0:0), leucyl-phenylalanine, 5,6-dihydrouracil 1, octahydro-2h-1-benzopyran-2-one, d-mannose, l-gulonolactone, adenine, cholestane-3β,5α,6β-triol, and lysope(0:0/22:1(13z)), all of which were relatively high in the SCB group (Figure 2F). In a paired comparison between the SCB and SCR groups, the top 10 DMs were leucyl-phenylalanine, lysope(0:0/14:0), lysopc(16:1(9z)/0:0), cystine, n-acetyl-l-aspartic acid, stearic acid, 5′-methylthioadenosine, pc(20:4(8z,11z,14z,17z)/p-18:0), hypoxanthine, and sphinganine (Figure 2G), of which 5 metabolites, including leucyl-phenylalanine, cystine, stearic acid, 5′-methylthioadenosine, and pc(20:4(8z,11z,14z,17z)/p-18:0), were relatively high in the SCB group. 

### 3.3. Comparison between Egg Yolk Metabolites

As mentioned previously, there were relatively distinct differences in the composition of yolks between the domesticated and mallard ducks, as well as some differences between the boiled and raw yolks. Therefore, we further assessed the relationships between the relative contents of metabolites in different groups with different sample characteristics.

The correlation between the yolk metabolites of domesticated ducks (Shaoxing ducks) and mallards (Fenghua teals and captive mallards) was analyzed to understand the changes in yolk metabolites during duck domestication. Most metabolites in the domesticated and mallard duck yolks remained linearly correlated as illustrated in Figure 3A–D. The correlation coefficient r = 0.9909 (*p* < 0.0001) obtained in the metabolic analysis of the raw egg yolk samples from Shaoxing ducks and Fenghua teals revealed that domestication increased the relative concentration of genistein and decreased the concentrations of 5-[2h-pyrrol-4-(3h)-ylidenemethyl]-2-furanmethanol, harmalan, and 2-propenyl 2-aminobenzoate (Figure 3A). Similar results were obtained in the correlation analysis of the yolk metabolites in the boiled samples from Shaoxing ducks and Fenghua teals with a correlation coefficient of r = 0.9778 (*p* < 0.0001) in addition to a decrease in pyrrolidine (Figure 3C). The correlation coefficients obtained in the correlation analyses of metabolites in the raw and boiled egg yolks between Shaoxing ducks and captive mallards were 0.9913 (*p* < 0.0001) and 0.9928 (*p* < 0.0001), respectively (Figure 3B,D). In the raw samples, domestication increased the content of 2-acetylpyrrolidine while decreasing the contents of 3-amino-2-piperidone, 3-methylglutarylcarnitine, harmalan, 3′-hydroxy-e,e-caroten-3-one, (3s, 3′r,5r,6r)-7′,8′-didehydro-3,6-epoxy-5,6-dihydro-beta,beta-carotene-3′,5-diol, and lutein (Figure 3B). Similarly, domestication increased the content of 2-acetylpyrrolidine but decreased the contents of n-acetyl-l-aspartic acid, 5-[2h-pyrrol-4-(3h)-ylidenemethyl]-2-furanmethanol, 3-amino-2-piperidone, harmalan, and cucurbitaxanthin in the boiled samples (Figure 3D).

We analyzed the correlation between the metabolites in the boiled and raw samples of egg yolks from Fenghua teals, captive mallards, and Shaoxing ducks to evaluate the effect of boiling on the yolk metabolites. The results demonstrated that the changes in the metabolite contents were not so obvious after boiling (Figure 3E–G). The correlation analysis between metabolites in the boiled and raw yolks of Fenghua teals yielded a correlation coefficient r = 0.9933 (*p* < 0.0001) and showed a relatively low concentration of picolinic acid in the boiled yolks (Figure 3E). The correlation coefficient between the metabolites in the boiled and raw yolks of captive mallards was 0.9964 (*p* < 0.0001) and showed a relatively low concentration of pc(18:1(11z)/14:0) in the boiled yolks (Figure 3F). Figure 3G shows a coefficient of 0.9923 (*p* < 0.0001) based on the correlation analysis of metabolites in the boiled and raw yolk samples from Shaoxing ducks. The relative content of hypoletin 8-gentiobioside was higher in the boiled yolks than in the raw yolks, and there was a relatively low concentration of pc(18:1(11z)/14:0) found in the boiled yolks.

### 3.4. Metabolic Response to Domestication and Boiling

We examined the overlap of DMs in different comparisons that included paired comparisons of SCR vs. FTR, SCR vs. CMR, SCB vs. FTB, and SCB vs. CMB between the domesticated ducks and mallards as well as pairwise comparisons of FTB vs. FTR, CMB vs. CMR, and SCB vs. SCR between the boiled and raw samples to determine the effects of domestication and boiling on metabolites. The results are presented in Figure 4 and Table 1 and Table 2.

In the four pairwise comparisons of the domesticated and mallard ducks, 48 distinct metabolites that included conduritol b epoxide 2, dimethylethanolamine, beta-doradecin, monoolein, and diphenylamine overlapped and were significantly altered by domestication (Figure 4A; Table 1). Furthermore, the characteristics of 48 overlapping differential metabolites were evaluated via HCA (Figure 4B). In the wild duck clade, the largest HCA clusters distinguished mallards from domestic ducks, while the second-largest clusters distinguished Fenghua teals from captive mallards. The results showed that the concentrations of prolyl-gamma-glutamate, monoolein, emodin, daidzein, genistein, glycitein, apigenin 7-sulfate, glycocholic acid, and conduritol b epoxide 2 were high in the domesticated duck egg yolks. Meanwhile, the concentrations of dimethylethanolamine, 1h-pyrrole-2-carboxaldehyde, harmalan, mannitol, imidazole acetol-phosphate, ganodosterone, and diphenylamine were relatively high in the wild duck egg yolks (Figure 4B). In the three paired comparisons between the boiled and raw egg yolks, four differential metabolites; namely, cystine, leucyl-phenylalanine, isoleucyl-leucine, and 5′-methylthioadenosine, overlapped and showed concentrations that were significantly altered by boiling (Figure 4C; Table 2). HCA divided the samples into raw and boiled yolk clades, and all of these overlapping differential metabolites were higher in the boiled samples than in the raw ones (Figure 4D).

### 3.5. Correlation Analysis of Differential Metabolites

To determine the correlation of metabolites induced by domestication or boiling, a metabolite–metabolite correlation analysis among the differential metabolites was conducted by applying Spearman’s correlation coefficient. The correlation analysis of the 48 overlapping differential metabolites in the paired comparisons between the domesticated and wild ducks demonstrated that 2304 correlations were analyzed, of which 2244 represented significant correlations (*p* < 0.05). Among these significant correlations, 1556 were identified as positive and 688 were negative (Figure 5A). A total of 16 correlations were evaluated for the 4 overlapping DMs identified via pairwise comparisons of the boiling and raw samples; the results revealed that they were all statistically positive (*p* < 0.01) (Figure 5B).

## 4. Discussion

Metabolomics involves extensive simultaneous analyses of many metabolites to generate a metabolic profile [21,22]. Changes in metabolite levels are generally caused by genetic and environmental factors [23], disease, microbiota, drugs, toxins, and lifestyle-associated cellular and systemic variations [24,25,26,27]. Analytical metabolomics platforms can detect hundreds of metabolites in complex biological samples and monitor changes in these metabolites [28,29,30,31]. However, due to the physicochemical diversity of these metabolites, none of the currently available analytical tools can identify the entire metabolome of a sample. Therefore, combining different analytical techniques may be an excellent strategy for comprehensively analyzing cellular metabolomes [22,32]. Nevertheless, most metabolomics has been performed using single analytical methods. This study examined the metabolic profiles of raw and boiled egg yolk samples from Fenghua teals, captive mallards, and Shaoxing ducks by combining untargeted LC-MS and GC-MS techniques. The HCA results showed that the domesticated and mallard ducks clustered separately, which was confirmed by the OPLS-DA scatter plots. Pairwise comparisons were used to evaluate the differential metabolites, and the significance of the difference was determined using a *t*-test. The metabolites with VIP > 1 and *p* < 0.05 were selected as the differential metabolites for each paired comparison [33,34]. We found relatively significant differences in the composition of yolks between domesticated and mallard ducks but minor differences between boiled and raw egg yolks, which suggested that domestication had a greater effect on the yolk metabolites than boiling. This study used metabolomics to demonstrate that key egg characteristics such as metabolites in the yolk may be affected by genetic and environmental factors and that genetics plays a significant role in these changes.

An analysis of the major differential metabolites in the yolks of the domestic and mallard ducks revealed that domestication decreased the expression of various metabolites such as 2-hydroxymyristic acid, mannitol, undecanoic acid, 2-pyrrolidinone, pipecolic acid, alloxanthin, diphenylamine, pyrrolidine, lysopc(18:2(9z,12z)), and beta-doradecin but increased the levels of metabolites such as daidzein and glycitein. A previous study on the antibacterial activity of phytochemically characterized extracts against some phytopathogenic bacteria suggested that 2-hydroxymyristic acid might have some antibacterial effect against plant bacterial pathogens [35]. Mannitol, a bacterial metabolite known as polyalcohol, has been reported to activate latent persistent cells inside bacterial biofilms. In a study that evaluated injectable mixtures of mannitol, chitosan, and polyethylene glycol for delivery of antibiotics and mannitol to eradicate staphylococcal biofilms, it was found that the mixtures could be clinically loaded with the clinician’s choice of antibiotics as adjunctive therapy for the prevention and treatment of musculoskeletal infections [36]. Persistent cell and pathogen biofilms are important in developing chronic infectious diseases. Meanwhile, fatty acids may be promising antipersister or antibiofilm agents due to their antimicrobial activities. Jin et al. (2021) showed that medium-chain saturated fatty acids such as undecanoic acid, lauric acid, and n-tridecanoic acid could be used as antipersistent agents and tepid antibiotic agents in the treatment of bacterial infections [37]. Thangam et al. (2013) proved that 2-pyrrolidone-rich parts of cruciferous plants exhibited antioxidant and in vitro anticancer activities [38]. Pipecolic acid is a nonproteinaceous product of lysine catabolism, an important regulator of immunity in plants and humans [39]. A study reported that alloxanthin and diatoxanthin from *Halocynthia roretzi* exhibited suppressive effects on proinflammatory cytokines and enzymes induced by LPS in RAW264.7 cells [40]. Diphenylamine is a common structure of nonsteroidal anti-inflammatory drugs (NSAIDs) that uncouples mitochondrial oxidative phosphorylation, thereby leading to decreased hepatocyte ATP content and hepatocyte injury. Studies on the structure and activity of pyrrolidine have shown that it has important antibacterial properties [41,42]. A previous study of the protective effect of quercetin against cadmium toxicity found that the concentration of lysopc(18:2(9z,12z)) in the cadmium-treated group increased. Additionally, its intensity in the high-dose quercetin plus a cadmium-treated group showed a reversible change compared with the cadmium-treated group, which suggested its protective effect against cadmium toxicity by regulating lysopc(18:2(9Z, 12Z)) [43]. The methanol extract (DBME) suppressed intracellular ROS formation and downregulated LPS-induced iNOS, COX-2, and TNF expression, which suggested its potential as an anti-inflammatory agent. A GC-MS analysis revealed the presence of hexadecanoic acid, tetradecanoic acid, hexadecen-1-ol, trans-9, 1-tetradecanol, fatty acid derivatives, and beta-doradecin in DBME, thereby suggesting its potential anti-inflammatory activities [44]. These metabolites are linked to the relatively high immunity, anti-inflammatory, antibacterial, and antioxidant activities in wild ducks compared with domestic ducks, which may be crucial for mallards to adapt to their harsh living environment. In addition, daidzein and glycitein, which are the main components of soy isoflavones, were significantly higher in the yolks of domesticated ducks than in those of wild ducks. Soy isoflavones are flavonoids, which are a class of secondary metabolites formed during the growth of soybeans that are typically biologically active. Soy isoflavones are also known as phytoestrogens because they are extracted from plants and have a structure similar to that of estrogen. The findings suggested that the higher egg production in domesticated ducks compared with the wild ducks may be attributed to the soy isoflavone intake by the former, which requires further validation.

Compared to the raw samples, the levels of carboxylic acids and derivatives (4-guanidinobutanoic acid, isoleucyl-leucine, leucyl-phenylalanine, and cystine), fatty acids and conjugates (pe(18:0/22:6(4z,7z,10z,13z,16z,19z)), lysopc(14:0/0:0), and lysope(0:0/22:1(13z)), stearic acid, pc(20:4(8z,11z,14z,17z)/p-18:0)), nucleosides, nucleotides and derivatives (adenine and 5′-methylthioadenosine), carbohydrates and carbohydrate conjugates (d-mannose), carbonyl compounds (3-hydroxy-3-(3,4-dihydroxy-4-methylpentanoyl)-5-(3-methylbutyl)-1,2,4-cyclopentanetrione), lactones (l-gulonolactone) and benzopyrans (octahydro-2h-1-benzopyran-2-one and cholestane-3,5,6-triol,(3beta,5alpha,6beta)-) increased in the boiled egg yolk; while phospholipids such as (pc(18:2(9z,12z)/15:0), pc(16:1(9z)/15:0), lysope(0:0/14:0), and lysopc(16:1(9z)/0:0)) and amino acids such as n-acetyl-l-aspartic acid and hypoxanthine decreased. Changes in metabolite contents may be linked to an improvement in the flavor of cooked egg yolks. For example, the production of hypoxanthine, a compound that contributes to an unpleasant taste, was inhibited by boiling [45].

The correlation analysis of the egg yolk metabolites between the domesticated and mallard ducks showed that the concentration of genistein was higher in the domesticated ducks than in the Fenghua teals, whereas the opposite was observed for the compound harmalan. Genistein is an isoflavone that is widely distributed in the legume family. Mammalian genistein is similar to daidzein and glycitein, which have exhibited estrogen-like functions [46]. The increase in these molecules in Shaoxing ducks may likewise be responsible for the improvement in the reproductive capacity of domestic ducks. Harmalan belongs to aromatic β-carboline alkaloids, which are found in traditional medicinal plants [47], bovine lungs [48], and rat organs [49]. Their anti-inflammatory activity has been established [50]. The egg yolks of mallard ducks had a relatively high concentration of harmalan, which might explain why mallards are more disease-resistant than domestic ducks. Furthermore, we used raw and boiled samples to characterize the differences in the thermal properties of the egg yolks. The results demonstrated that boiling reduced the content of picolinic acid in the Fenghua teal egg yolks compared to the raw ones. The most notable reactions were oxidation, reduction, hydrolysis, and aromatic ring destruction. It was hypothesized that heating treatment may expedite these processes, thereby decreasing the picolinic acid content in egg yolks [51]. Additionally, boiling reduced the content of pc(18:1(11z)/14:0) in the yolks of captive mallards and Shaoxing ducks. Lipids—particularly phospholipids—are important precursors to flavor compounds in meat. Phospholipids are rich in unsaturated fatty acids, and their products directly influence the composition of volatile flavor compounds, thereby contributing to the meat flavor [52]. A previous study that demonstrated the possible conversion of glycerophosphorylcholine (pc(18:1(11z)/14:0)) into flavor compounds by heating resulted in cooked meat with enhanced sour, sweet, bitter, salty, and fresh flavors [53]. Correlation analyses between boiled and raw egg yolks of the three duck breeds showed that boiling-induced changes in the yolk metabolites were limited. These results reaffirmed that domestication causes greater changes in yolk metabolites than boiling.

In this study, 48 metabolites overlapped in four paired comparisons of the impact of domestication on egg yolk metabolites. Nine metabolites that included monoolein, emodin, daidzein, genistein, glycitein, and glycocholic acid were shown to be significantly higher in domesticated ducks versus wild ducks. Monoolein, which is one of the most important lipids, has multiple applications in drug delivery, emulsion stabilization, and protein crystallization [54], while emodin has been associated with weight loss effects due to its lipid-lowering properties and capability to alleviate CCL4-induced steatosis [55]. Meanwhile, the antiobesity effects of genistein and glycitein have been reported through their role in activating adipose tissue metabolism [56], whereas glycocholic acid (also known as cholecylglycine) is a crystalline bile acid that is reportedly involved in fat emulsification [57]. Our findings implied that these metabolites may be associated with the domestication of the Shaoxing duck, which may be involved in lipid metabolism. A higher level of 39 overlapping differential metabolites that included dimethylethanolamine, nornicotine, linoleic acid, diphenylamine, proline betaine, alloxanthin, and resolvin d1 in mallard ducks compared with domestic ducks was demonstrated through a set of four pairwise comparisons. Dimethylethanolamine (N,N-dimethylethanolamine) is a colorless, volatile liquid with an unpleasant ammonia odor that may be associated with the heavy, earthy stench of wild duck eggs. Grebenstein et al. (2022) showed that short-term daily administration of minor tobacco alkaloids (nornicotine, anatabine, and anabasine) produced a positive effect on weight loss by reducing body fat and food intake and increasing physical activity [58]. Linoleic acid is known to be an important fatty acid for both humans and animals. Conjugated linoleic acid refers to a mixture of all stereoisomers and positional isomers of linoleic acid, which is vital in preventing obesity and its comorbidities (including type 2 diabetes and cardiovascular disease) by reducing blood cholesterol in humans [59]. The relatively high contents of nornicotine and linoleic acid in the wild duck egg yolks may be associated with a lower body fat deposition in the wild ducks than in the domestic ducks. A series of diphenylamine containing 1,2,4-triazoles have shown potent activity against *Mycobacterium tuberculosis* H37Rv strain [60]; proline betaine is an alkaloid isolated from motherwort that has been shown to have antioxidant, anti-inflammatory, anticancer, and cardioprotective properties [61]; alloxanthin, a major CAC carotenoid, showed anti-inflammatory effects by inhibiting the overexpression of cyclooxygenase-2 and nitric oxide synthase mRNA in RAW264.7 cells induced by LPS [40]; and resolvin D1 is endogenously produced from ω3 polyunsaturated fatty acids. As specialized proresolving mediators, these compounds can inhibit the expression of inflammatory cytokines and enhance cellular protection against oxidative stress by stimulating lipoxin A4 receptor/formyl peptide receptor 2 to promote the production of antioxidant proteins [62]. Such roles suggest the potential anti-inflammatory and antioxidant activities of these compounds that may exert antidisease effects in wild ducks. Interestingly, a higher level of 1h-pyrrole-2-carboxaldehyde, a pheromone found in the urine of female pandas that elicits behavioral and physiological responses in male pandas, was found in mallard egg yolks [63]. Thus, this compound maybe a signal compound released by female mallard ducks to increase mating prospects in the wild.

Additionally, four pairwise comparisons between the domestic and mallard ducks revealed 48 overlapping differential metabolites that might have emerged during domestication. Of the 48 identified differential metabolites, 9 were upregulated and 39 were downregulated. A correlation heatmap was created using Spearman’s correlation coefficient to illustrate the correlations among these overlapping differential metabolites. Except for a negative correlation between 3-methylglutarylcarnitine and pc(20:1(11z)/15:0), there were positive correlations among 39 metabolites reduced by domestication. Among these, n-acetyl-d-glucosamine was significantly positively correlated with 2′-o-methyladenosine (*p* < 0.01), which was similar to the latter two metabolites. N-acetyl-d-glucosamine is one of nature’s most abundant amino sugars and an essential constituent of many polysaccharides and glycoconjugates, which have been associated with various human diseases [64,65]. Additionally, 2′-O-methyladenosine has been associated with plant stress tolerance and immune response [66]. Our findings suggested the possible positive regulations between these resistance-related molecules. Similarly, there were positive correlations between any two of the nine metabolites upregulated by domestication as well as strong positive correlations between lipid regulatory molecules such as glycitein, genistein, daidzein, and emodin (*p* < 0.01). However, negative correlations were found between the metabolites that were upregulated and downregulated by domestication. Furthermore, we conducted a correlation analysis of the four overlapping differential metabolites caused by boiling and found highly significant positive correlations among them (*p* < 0.01) with strong correlations (r > 0.7) between any two of cystine, leucine-phenylalanine, and isoleucyl-leucine. The findings suggested that proteins or peptides in duck egg yolk may include these amino acids or dipeptides, which are released by hydrolysis during heating.

## 5. Conclusions

We found that domestication increased the metabolites related to lipid metabolism and decreased the compounds involved in stress resistance in the duck egg yolks. Boiling increased the contents of several amino acids that included cystine, leucyl-phenylalanine and isoleucyl-leucine. The results also demonstrated that domestication produced a noticeable impact on the metabolites of the duck egg yolks compared to boiling. Our findings implied that metabolomics can be useful for examining changes in duck egg yolk composition in response to domestication and boiling. This study resulted in useful knowledge of the composition of duck egg yolks and the associated influencing factors.

## Figures and Tables

**Figure 1 metabolites-13-00135-f001:**
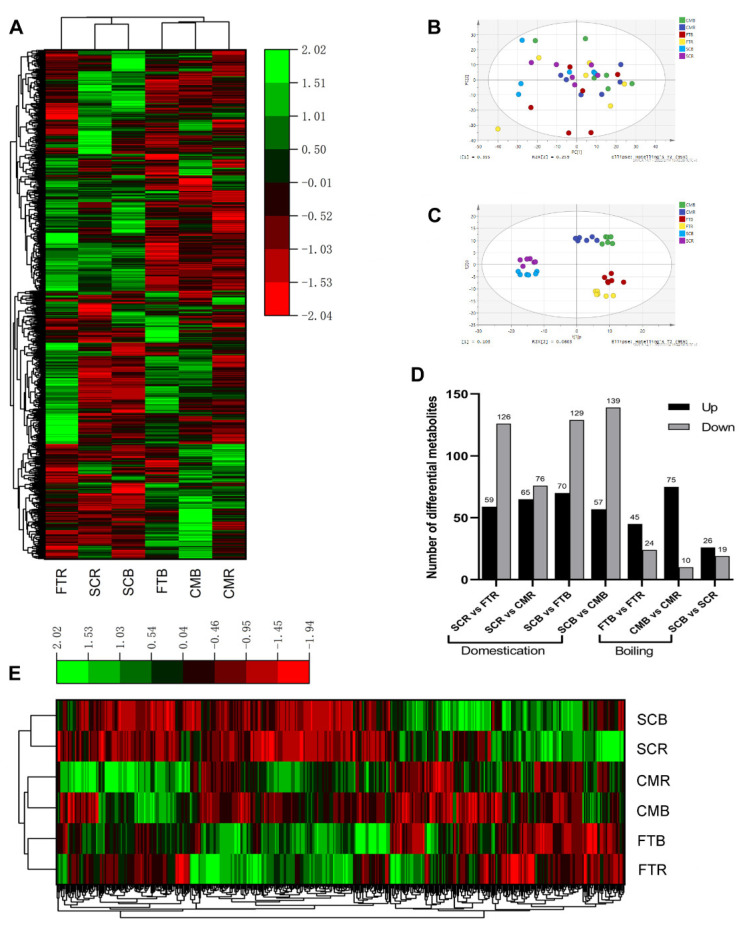
Metabolic profiles of yolk samples from six groups of duck eggs. (**A**) Heatmap of all identified metabolites. (**B**) Principal component analysis (PCA) score plots for all samples. (**C**) Orthogonal partial-least-squares discriminant analysis (OPLS-DA) score plots of the metabolomic data from six groups. (**D**) The number of differential metabolites in pairwise comparisons. (**E**) Hierarchical cluster analysis (HCA) of the differential metabolites identified in seven pairwise comparisons based on the normalized average abundance of the metabolomic profiles.

**Figure 2 metabolites-13-00135-f002:**
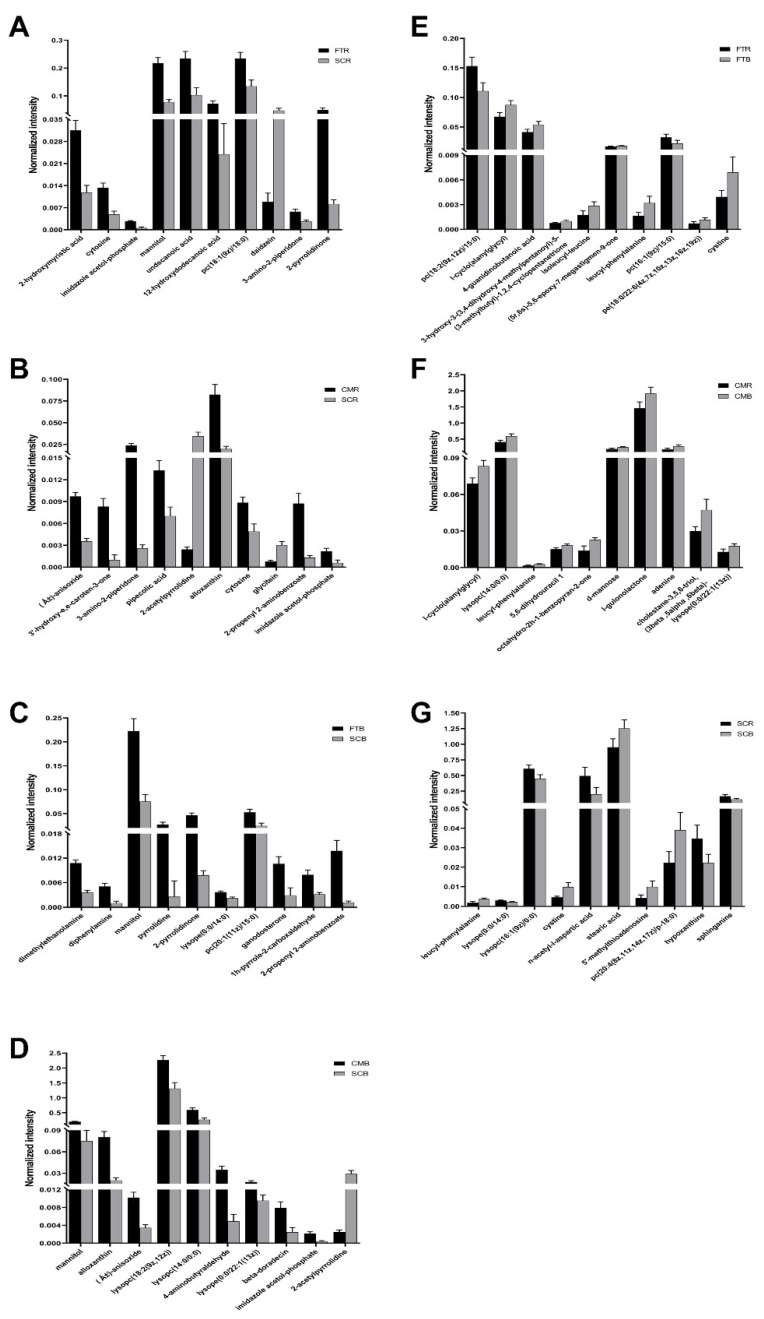
Top 10 differential metabolites based on the lowest *p*-values between: (**A**) Shaoxing duck and Fenghua teal yolks (raw samples); (**B**) Shaoxing duck and captive mallard yolks (raw samples); (**C**) Shaoxing duck and Fenghua teal yolks (boiled samples); (**D**) Shaoxing duck and captive mallard yolks (boiled samples); (**E**) raw and boiled egg yolk samples of Fenghua teals; (**F**) raw and boiled captive mallard egg yolk samples; (**G**) raw and boiled egg yolk samples of Shaoxing ducks. The *X*-axis indicates the metabolites. The *Y*-axis represents the normalized area for the metabolites. The plot shows the means with SEMs.

**Figure 3 metabolites-13-00135-f003:**
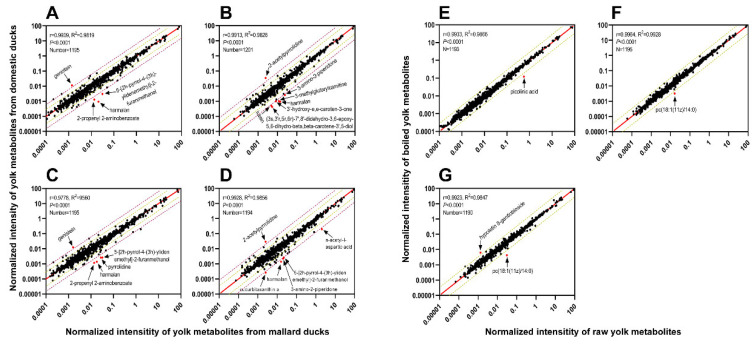
Correlation of metabolite concentrations in egg yolk samples between domestic and mallard ducks (**A**–**D**). (**A**) Correlation between Shaoxing duck and Fenghua teal raw yolk samples; (**B**) correlation between Shaoxing duck and captive mallard raw yolk samples; (**C**) correlation of boiled egg yolks between Shaoxing ducks and Fenghua teals; (**D**) correlation of boiled egg yolks between Shaoxing ducks and captive mallards. The horizontal and vertical axes indicate log scales of yolk metabolite concentrations from mallard and domesticated ducks, respectively. Correlation of metabolite concentrations between raw and boiled egg yolks (**E**–**G**). (**E**) Egg yolks of Fenghua teals; (**F**) egg yolks of captive mallards; (**G**) egg yolks of Shaoxing ducks. The horizontal and vertical axes indicate log scales of metabolite concentrations of raw and boiled duck egg yolks, respectively.

**Figure 4 metabolites-13-00135-f004:**
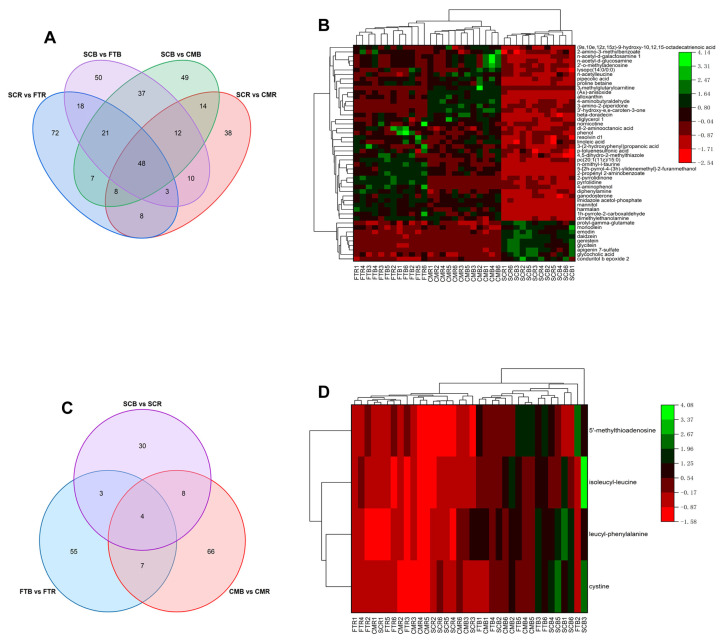
(**A**) Venn diagram showing numbers of differential metabolites involved in domestication based on paired comparisons of SCR vs. FTR, SCR vs. CMR, SCB vs. FTB, and SCB vs. CMB datasets. (**B**) Changes in patterns of the differential metabolites induced by domestication visualized in a heatmap with HCA. (**C**) Venn diagram showing numbers of differential metabolites induced by boiling based on paired comparisons of FTB vs. FTR, CMB vs. CMR, and SCB vs. SCR datasets. (**D**) HCA of the overlapping differential metabolites identified in three pairwise comparisons between boiled and raw egg yolk samples.

**Figure 5 metabolites-13-00135-f005:**
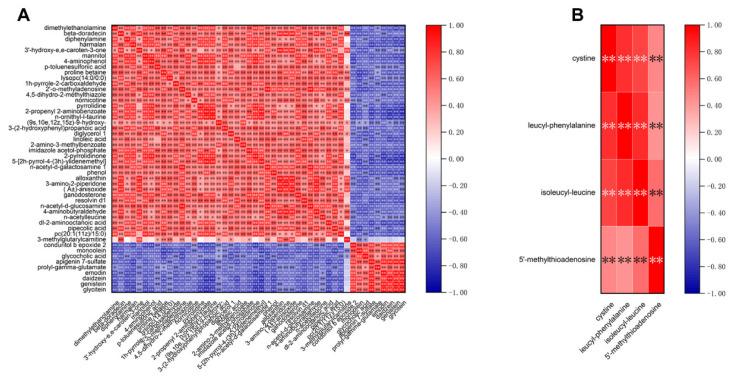
Correlation heatmaps of differential metabolites induced by domestication (**A**) and boiling (**B**). Each square indicates r (Spearman’s correlation coefficient value for a pair of metabolites). Positive (0 < r ≤1) correlations are shown in red, and negative (−1 < r < 0) correlations are in blue (* 0.01 ≤ *p* < 0.05, ** *p* < 0.01).

**Table 1 metabolites-13-00135-t001:** Metabolites with domestication-induced changes.

Metabolites	Class ^1^	Relative Area (Yolk)						SEM	*p*-Value			Trend ^2^
		Fenghua Teal		Captive Mallard		Shaoxing Duck						
		Wild		Wild		Domestic						
		Raw	Boiled	Raw	Boiled	Raw	Boiled		Domestication	Boiling	Domestication × Boiling	
Conduritol b epoxide 2	/	0.0039	0.0035	0.0047	0.0046	0.0110	0.0096	0.0006	<0.001	0.187	0.393	Up
Dimethylethanolamine	Amines	0.0096	0.0108	0.0091	0.0082	0.0044	0.0035	0.0005	<0.001	0.560	0.477	Down
Beta-doradecin	Prenol lipids	0.0063	0.0056	0.0079	0.0079	0.0025	0.0025	0.0004	<0.001	0.733	0.761	Down
Monoolein	Glycerolipids	0.0015	0.0013	0.0014	0.0013	0.0022	0.0024	0.0001	<0.001	0.900	0.161	Up
Diphenylamine	Benzene and substituted derivatives	0.0055	0.0051	0.0031	0.0030	0.0012	0.0010	0.0003	<0.001	0.618	0.886	Down
Harmalan	Harmala alkaloids	0.0206	0.0181	0.0137	0.0158	0.0009	0.0014	0.0015	<0.001	0.951	0.832	Down
3′-Hydroxy-e,e-caroten-3-one	Prenol lipids	0.0022	0.0025	0.0083	0.0081	0.0010	0.0013	0.0005	<0.001	0.890	0.886	Down
Mannitol	Organooxygen compounds	0.2176	0.2227	0.1794	0.1986	0.0783	0.0753	0.0109	<0.001	0.576	0.353	Down
Glycocholic acid	Steroids and steroid derivatives	0.0009	0.0008	0.0006	0.0006	0.0025	0.0035	0.0002	<0.001	0.114	0.091	Up
4-aminophenol	Benzene and substituted derivatives	0.0057	0.0065	0.0028	0.0028	0.0020	0.0017	0.0004	<0.001	0.910	0.565	Down
p-Toluenesulfonic acid	Organic sulfuric acids and derivatives	0.4382	0.4090	0.4130	0.4116	0.3208	0.2970	0.0129	<0.001	0.353	0.838	Down
Proline betaine	Carboxylic acids and derivatives	0.6744	0.7447	0.7691	0.8050	0.2035	0.1280	0.0548	<0.001	0.862	0.320	Down
Lysopc(14:0/0:0)	Glycerophospholipids	0.4477	0.4533	0.4105	0.5938	0.3419	0.2679	0.0210	<0.001	0.745	0.011	Down
Apigenin 7-sulfate	Flavonoids	0.0002	0.0002	0.0003	0.0002	0.0014	0.0013	0.0001	<0.001	0.677	0.733	Up
1h-Pyrrole-2-carboxaldehyde	Organooxygen compounds	0.0088	0.0079	0.0074	0.0079	0.0029	0.0032	0.0005	<0.001	0.995	0.697	Down
2′-o-Methyladenosine	Purine nucleosides	0.0156	0.0257	0.0215	0.0427	0.0083	0.0062	0.0026	<0.001	0.088	0.027	Down
3-Methyl-glutarylcarnitine	Fatty acyls	0.0014	0.0013	0.0118	0.0170	0.0011	0.0015	0.0012	0.008	0.535	0.646	Down
4,5-Dihydro-2-methylthiazole	Azolines	0.1286	0.1348	0.1234	0.1276	0.1145	0.1097	0.0018	<0.001	0.940	0.068	Down
Nornicotine	Pyridines and derivatives	0.0060	0.0038	0.0045	0.0048	0.0025	0.0025	0.0003	<0.001	0.321	0.373	Down
Pyrrolidine	Pyrrolidines	0.0259	0.0273	0.0078	0.0073	0.0041	0.0022	0.0018	<0.001	0.820	0.718	Down
2-Propenyl 2-aminobenzoate	Benzene and substituted derivatives	0.0135	0.0138	0.0087	0.0086	0.0013	0.0012	0.0009	<0.001	0.976	0.937	Down
n-Ornithyl-l-taurine	Carboxylic acids and derivatives	0.0261	0.0278	0.0163	0.0168	0.0083	0.0080	0.0014	<0.001	0.856	0.719	Down
(9s,10e,12z,15z)-9-Hydroxy-10,12,15-octadecatrienoic acid	Lineolic acids and derivatives	0.0179	0.0194	0.0183	0.0196	0.0144	0.0160	0.0005	<0.001	0.066	0.904	Down
3-(2-Hydroxyphenyl)propanoic acid	Phenylpropanoic acids	0.1464	0.1264	0.0822	0.0840	0.0241	0.0283	0.0109	<0.001	0.897	0.732	Down
Prolyl-gamma-glutamate	Carboxylic acids and derivatives	0.0033	0.0027	0.0044	0.0030	0.0069	0.0060	0.0003	<0.001	0.056	0.923	Up
Emodin	Anthracenes	0.0177	0.0213	0.0235	0.0311	0.0725	0.0838	0.0048	<0.001	0.056	0.512	Up
Diglycerol 1	/	0.0040	0.0039	0.0045	0.0041	0.0032	0.0032	0.0001	<0.001	0.239	0.447	Down
Linoleic acid	Fatty acyls	18.9801	17.9344	16.0182	20.3844	12.5212	13.2524	0.6534	<0.001	0.264	0.661	Down
2-Amino-3-methylbenzoate	Benzene and substituted derivatives	0.0115	0.0119	0.0110	0.0126	0.0096	0.0101	0.0002	<0.001	0.062	0.447	Down
Metabolites												
Imidazole acetol-phosphate	Organic phosphoric acids and derivatives	0.0027	0.0021	0.0022	0.0022	0.0006	0.0004	0.0002	<0.001	0.106	0.513	Down
2-Pyrrolidinone	Pyrrolidines	0.0494	0.0464	0.0158	0.0140	0.0082	0.0078	0.0030	<0.001	0.786	0.851	Down
5-[2h-Pyrrol-4-(3h)-ylidenemethyl]-2-furanmethanol	Heteroaromatic compounds	0.0312	0.0322	0.0207	0.0216	0.0030	0.0026	0.0022	<0.001	0.922	0.793	Down
n-Acetyl-d-galactosamine 1	/	0.0137	0.0174	0.0166	0.0266	0.0078	0.0065	0.0014	<0.001	0.174	0.050	Down
Phenol	Phenols	0.0027	0.0042	0.0025	0.0030	0.0021	0.0019	0.0002	0.001	0.258	0.057	Down
Alloxanthin	Prenol lipids	0.0480	0.0501	0.0825	0.0804	0.0198	0.0200	0.0044	<0.001	0.984	0.990	Down
3-Amino-2-piperidone	Carboxylic acids and derivatives	0.0057	0.0056	0.0240	0.0202	0.0026	0.0022	0.0015	<0.001	0.660	0.764	Down
Daidzein	Isoflavonoids	0.0089	0.0082	0.0101	0.0098	0.0479	0.0479	0.0032	<0.001	0.887	0.924	Up
(Â±)-anisoxide	/	0.0064	0.0069	0.0097	0.0102	0.0035	0.0035	0.0005	<0.001	0.748	0.654	Down
Genistein	Isoflavonoids	0.0014	0.0016	0.0019	0.0020	0.0113	0.0125	0.0009	<0.001	0.266	0.365	Up
Ganodosterone	Steroids and steroid derivatives	0.0114	0.0106	0.0097	0.0103	0.0041	0.0029	0.0006	<0.001	0.374	0.449	Down
Resolvin d1	Fatty acyls	0.3247	0.2843	0.2387	0.2931	0.1821	0.1774	0.0114	<0.001	0.946	0.732	Down
Glycitein	Isoflavonoids	0.0005	0.0006	0.0008	0.0007	0.0030	0.0027	0.0002	<0.001	0.307	0.269	Up
n-Acetyl-d-glucosamine	Organooxygen compounds	0.0187	0.0249	0.0236	0.0398	0.0100	0.0089	0.0022	<0.001	0.138	0.075	Down
4-Aminobutyraldehyde	Organooxygen compounds	0.0143	0.0123	0.0316	0.0347	0.0058	0.0049	0.0021	<0.001	0.957	0.834	Down
n-Acetylleucine	Carboxylic acids and derivatives	0.0300	0.0339	0.0378	0.0405	0.0218	0.0243	0.0015	<0.001	0.239	0.868	Down
dl-2-Aminooctanoic acid	Carboxylic acids and derivatives	0.0063	0.0075	0.0042	0.0056	0.0028	0.0026	0.0004	<0.001	0.444	0.267	Down
Pipecolic acid	Carboxylic acids and derivatives	0.0105	0.0126	0.0133	0.0148	0.0070	0.0074	0.0006	<0.001	0.174	0.373	Down
pc(20:1(11z)/15:0)	Glycerophospholipids	0.0458	0.0527	0.0314	0.0380	0.0242	0.0245	0.0020	<0.001	0.264	0.305	Down

^1^ Classification of substances in HMDB database. ^2^ “Up” indicates an increase in metabolite in domesticated duck yolks compared to wild duck yolks; “Down” indicates a decrease in metabolite in domesticated duck yolks compared to wild duck yolks.

**Table 2 metabolites-13-00135-t002:** Metabolites with boiling-induced changes.

Metabolites	Class ^1^	Relative Area (Yolk)						SEM	*p*-Value			Trend ^2^
		Fenghua Teal		Captive Mallard		Shaoxing Duck						
		Wild		Wild		Domestic						
		Raw	Boiled	Raw	Boiled	Raw	Boiled		Boiling	Domestication	Boiling × Domestication	
Cystine	Carboxylic acids and derivatives	0.0039	0.0069	0.0035	0.0062	0.0047	0.0098	0.0004	<0.001	<0.001	0.044	Up
Leucyl-phenylalanine	Carboxylic acids and derivatives	0.0017	0.0032	0.0016	0.0028	0.0018	0.0037	0.0002	<0.001	0.023	0.203	Up
Isoleucyl-leucine	Carboxylic acids and derivatives	0.0017	0.0029	0.0017	0.0030	0.0018	0.0035	0.0002	<0.001	0.254	0.333	Up
5′-Methylthioadenosine	5′-deoxyribonucleosides	0.0095	0.0156	0.0063	0.0124	0.0043	0.0099	0.0008	<0.001	0.001	0.846	Up

^1^ Classification of substances in HMDB database. ^2^ “Up” indicates an increase in metabolite in boiled duck egg yolks compared to raw duck egg yolks.

## Data Availability

The data presented in this study are available in the article and Appendix A.

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
