# Peer review of "Integration of LC-MS-Based and GC-MS-Based Metabolic Profiling to Reveal the Effects of Domestication and Boiling on the Composition of Duck Egg Yolks"

_metabolites, 2023, doi:10.3390/metabo13010135_

Round 1

Reviewer 1 Report

This manuscript investigates the influence of domestication and boiling on the composition of duck egg yolks through the integration of LC-MS and GC-MS based metabolic profiling in combination with chemometrics. It is an interesting study and the efforts to carry out this type of study that involves large sample size to increase the reliability of data are commendable. This manuscript is generally well written but there are a few comments for the authors to consider in revising the paper to improve its readability. Most references used are recent and up-to-date.

1) Please check the entire manuscript for some minor syntax errors.

2) Abstract: Line 23 GC-MS

Line 36-37 It would be better to rewrite the last sentence to conclude the impact of the significant findings.

3) Introduction: Lack of rationale of study/ hypothesis. More descriptions on domestication and boiling are needed to demonstrate the novelty of this study. 

4) Materials & methods: Is there any criteria for the selection of eggs? Eg grade/size? Weight of eggs are different (line 96-97), will this factor affect the composition? Is weight one of the criteria in consideration?

Avoid using acronyms or abbreviations directly without introducing the terms in full for its first use in the body text.

Line 138 sample were ground? Please double check.

5) Results: This study involves big data and multivariate analysis, would be good to explain and elaborate (present) the data, especially the heatmap, HCA and DMs.

Fig 2, give footnote for the horizontal white line

6) Discussion: Line 388-403 May consider shifting this part to introduction, as would be better to start this section with the elaboration of results obtained. 

The DMs are stated with their functions, but slightly lack of association back to the factors, i.e. domestication and boiling. How the domestication affect the metabolite? Feed, practices etc? How about boiling duration? Will it bring significant impact to the metabolic profiles? Discuss briefly.

7) Conclusion: Need to rewrite to highlight (summarize) the significant findings and if it answers the objective in this study. 

Reviewer 2 Report

I appreciate all the authors for taking the initiative to study egg yok metabolome, the study did seem complete and thorough.

I have the following minor comments for the authors:

Line 107, please check unit

Line 114: What's the internal standard?

Figure 3. Image quality needs to be improved or legends/data point labeling must be described separately

I would advise the authors to thoroughly check for minor errors and mistakes before resubmitting to the editor.

Reviewer 3 Report

This was a well conducted analysis of duck egg yolks which provides valuable information on the chemical makeup of duck egg yolks for future researchers. 

major critique

For the statistical analyses many tests were run. How were these corrected for multiple comparisons (I.E. False Discovery Rate)? For example when performing correlation tests there were 2304 comparisons tested and the resulting p-value was reported. A multiple comparison correction needs to be applied to these p-values. For example the Benjamini-Hochberg correction is most commonly used for metabolomics analyses. Raw p-values and Corrected p-values are usually both reported for these types of analyses. Please include an FDR statement or correction in the manuscript.

minor critiques

Could you specify the "internal standard mixture" mentioned in the methods section for LCMS? List of compounds or reference to the mixture?

In LCMS information-dependent acquisition how many ions were selected for fragmentation during each scan loop? 

The term "placed in the refrigerator at -40 °C" should be changed to something like "stored at -40C" since the word refrigerator is usually used for storage above freezing i.e. 4C.

In methods on metabolite identification please specify what criteria were used to identify metabolites. For example what was the MS/MS matching criteria was used to identify metabolites? Was it based only on MS1 accurate mass matching for LC-MS or also MS/MS matching? Were the matches manually inspected and what was done if one metabolite was matched to multiple MS features (duplicates)? 

Were internal standards used for anything during LCMS data analysis? Please specify or give information gained from the internal standards. How were the IS used to normalize for LCMS?

Would it be possible to add a column in Tables 1 and 2 to distinguish the direction of change? It is hard to read all the values to quickly distinguish between metabolites increased/decreased with domestication. 

Figure 4 A and C have a numeric code underneath the labels. As these aren't referenced or explained they could be removed from the figures. 

Figure 4B has some overlapping text making it impossible to read the metabolite names. Perhaps simplifying the name or some other method could avoid this.

The term "shown to be relatively high" on line 503 should be more accurate. Such as "shown to be significantly higher in domesticated versus wild"

In the discussion 1h-pyrrole-2-carboxaldehyde is mentioned as a panda pheromone, but in HMDB it is listed as a component of different plants. It seems more plausible that this chemical comes from the different diet of the wild duck as compared to a signaling pheromone. Perhaps consider this other possibility.
